# Causal structure of interacting Weyl fermions in condensed matter systems

Wei-Chi Chiu [1,15], Guoqing Chang [2,15] ✉, Gennevieve Macam[3,4], Ilya Belopolski[5,6], Shin-Ming Huang[4,7,8], Robert Markiewicz[1], Jia-Xin Yin[9], Zi-Jia Cheng[5], Chi-Cheng Lee [10], Tay-Rong Chang[7,11,12], Feng-Chuan Chuang[4,7,8], Su-Yang Xu [13], Hsin Lin [14], M. Zahid Hasan [5] & Arun Bansil [1]

The spacetime light cone is central to the definition of causality in the theory of relativity. Recently, links between relativistic and condensed matter physics have been uncovered, where relativistic particles can emerge as quasiparticles in the energy-momentum space of matter. Here, we unveil an energy-momentum analogue of the spacetime light cone by mapping time to energy, space to momentum, and the light cone to the Weyl cone. We show that two Weyl quasiparticles can only interact to open a global energy gap if they lie in each other's energy-momentum dispersion cones–analogous to two events that can only have a causal connection if they lie in each other's light cones. Moreover, we demonstrate that the causality of surface chiral modes in quantum matter is entangled with the causality of bulk Weyl fermions. Furthermore, we identify a unique quantum horizon region and an associated 'thick horizon' in the emergent causal structure.

The intriguing connections between high energy and condensed matter physics have led to a deeper understanding of quantum matter[1–14]. One such connection manifests itself in topological materials where relativistic particles can emerge as quasiparticles. A familiar example is the Weyl fermion, a massless spin-1/2 particle proposed in 1929, which has been realized in many condensed matter systems[10–13]. Weyl fermions have also attracted recent attention due to their unique quantum responses, such as the quantized circular photogalvanic effect[15–21]. Another frontier concerns strongly interacting systems which host unusual effects driven by the interplay of correlations, topology, and geometry[2–8].

Here, correlated Weyl semimetals provide a perfect platform for exploring interaction effects on single-particle physics. The separation of the individual Weyl nodes with opposite topological charges in momentum space makes it impossible to hybridize these nodes and produce a fully gapped insulating state without violating symmetries. In correlated systems, however, interactions can in principle open a global gap in the system with Weyl fermions. For instance, it has been reported that a Weyl semimetal can be gapped out into an axion insulator by the charge-density-wave (CDW) pairing interaction[22–25], although the general mechanism of this metal-insulator transition remains elusive. Even though Weyl fermions are rooted in quantum

[1]Department of Physics, Northeastern University, Boston, MA 02115, USA. [2]Division of Physics and Applied Physics, School of Physical and Mathematical Sciences, Nanyang Technological University, 21 Nanyang Link, 637371 Singapore, Singapore. [3]National Institute of Physics, University of the Philippines, Diliman, Quezon City 1101, Philippines. [4]Department of Physics, National Sun Yat-sen University, Kaohsiung 80424, Taiwan. [5]Laboratory for Topological Quantum Matter and Advanced Spectroscopy (B7), Department of Physics, Princeton University, Princeton, NJ 08544, USA. [6]RIKEN Center for Emergent Matter Science (CEMS), Wako, Saitama 351-0198, Japan. [7]Physics Division, National Center for Theoretical Sciences, Taipei 10617, Taiwan. [8]Center for Theoretical and Computational Physics, National Sun Yat-sen University, Kaohsiung 80424, Taiwan. [9]Department of Physics, Southern University of Science and Technology, Shenzhen, Guangdong 518055, China. [10]Department of Physics, Tamkang University, Tamsui, New Taipei 251301, Taiwan. [11]Department of Physics, National Cheng Kung University, Tainan, Taiwan. [12]Center for Quantum Frontiers of Research and Technology (QFort), Tainan, Taiwan. [13]Department of Chemistry and Chemical Biology, Harvard University, Cambridge, MA, USA. [14]Institute of Physics, Academia Sinica, Taipei 115201, Taiwan. [15]These authors contributed equally: Wei-Chi Chiu, Guoqing Chang. ✉e-mail: guoqing.chang@ntu.edu.sg

field theory, how causal physics[1] enters the interaction dynamics of Weyl semimetals remains unexplored. With this motivation, we discuss a topological phase transition mechanism for a CDW-correlated Weyl semimetal. We focus on a more likely scenario in real materials, wherein the CDW arises in a system where the Weyl nodes exist but are not caused by the Weyl nodes and hence it can possess a different periodicity (Fig. 1a).

## Results

We start from an inversion-symmetry breaking model with four Weyl quasiparticles in the first Brillouin zone (BZ):

$$\mathcal{H}(k) = A \sin k_x \sin k_z \sigma_0 + \left[ \cos(k_x + \theta) - \cos k_1 \right] \sigma_x \\ + \sin k_y \sigma_y + \left( 1 + \cos k_z - \cos k_y \right) \sigma_z, \tag{1}$$

where $A$ and $k_1$ are two constants with $k_1 \neq \pi$, and the $\sigma$ are the Pauli matrices. We first consider the case with $\theta = 0$ which preserves the time-reversal symmetry of the system and makes the Fermi-velocity of each Weyl quasiparticle roughly the same. The positions of four Weyl nodes with $\mp 1$ chirality are at $\mathbf{k}_W = \pm (k_1, 0, \pm \pi/2)$ with black (white) dots

representing positive (negative) chirality. The Weyl nodes of different chirality are at energies $E_W = \pm A \sin k_1$ (Fig. 1b).

To discuss the dynamics of interacting Weyl fermions, we consider the CDW instability as a quasi-one-dimensional Peierls instability such that there is only one unidirectional CDW Q-vector ($\mathbf{Q}_{CDW}$). For simplicity, but without losing generality, we fix $\mathbf{Q}_{CDW}$ and vary the separation between the Weyl nodes instead. We choose $\mathbf{Q}_{CDW} = (\pi, 0, 0)$ as a representative, which reflects the Peierls dimerization in a double supercell along the $x$-direction, see Methods section titled: CDW tight-binding model in real space.

When the CDW wavevector is equal to the momentum separation of the Weyl fermions ($\mathbf{Q}_{CDW} = \mathbf{k}_{W_1} - \mathbf{k}_{W_2}$) and the Weyl nodes lie at the same energy ($E_{W_1} = E_{W_2}$), our calculations show that the CDW interaction will gap out the Weyl fermions, which is consistent with previous work[24,25]. However, when the separation of the two Weyl nodes is not the source of the nesting vector of CDW ($\mathbf{Q}_{CDW} \neq \mathbf{k}_{W_1} - \mathbf{k}_{W_2}$), the whole system remains a Weyl semimetal, see Supplementary Fig. 1. This indicates that the relationship between the Q-vector and the separation of the Weyl nodes plays a key role in determining the topological phase transition in correlated Weyl semimetals.

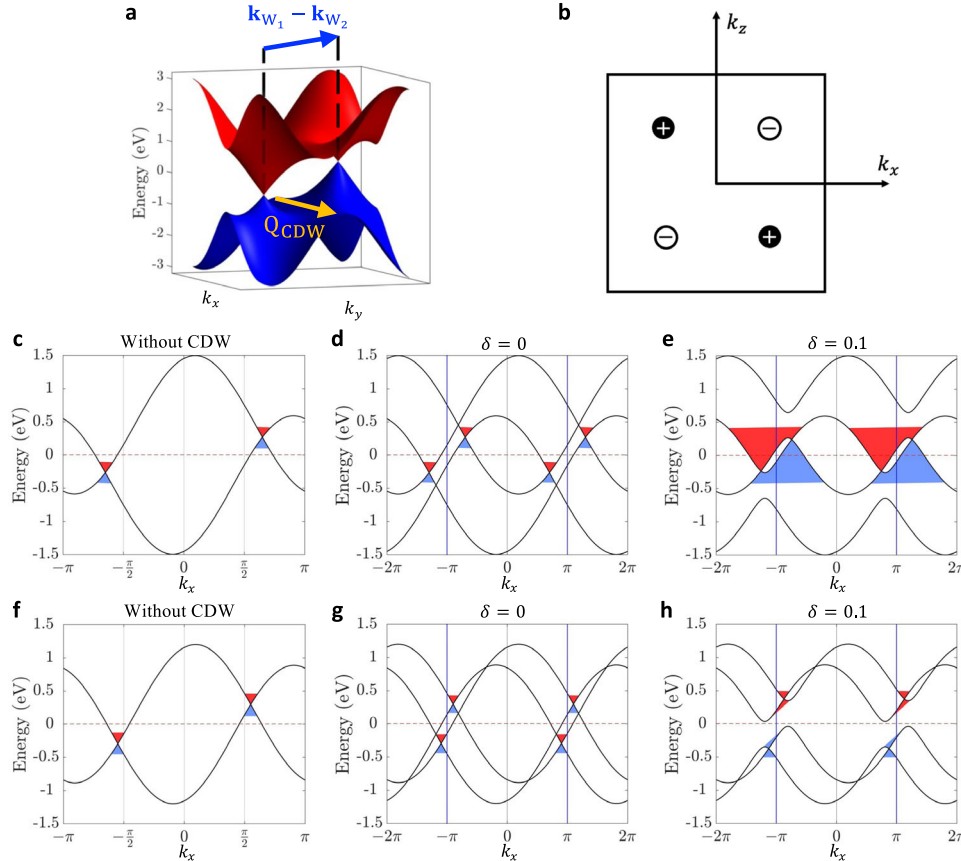

**Fig. 1 | CDW interactions in Weyl semimetals. a** Schematic illustration of two Weyl nodes with an energy difference and CDW Q-vector ($\mathbf{Q}_{CDW}$) is not equal to the separation of the two Weyl nodes ($\mathbf{k}_{W_1} - \mathbf{k}_{W_2}$). The yellow arrow represents $\mathbf{Q}_{CDW}$, and the blue arrow represents $\mathbf{k}_{W_1} - \mathbf{k}_{W_2}$. The red/blue structures represent the conduction/valence dispersion cones. The $k_x$, $k_y$, and $k_z$ represent the directions in momentum space. **b** Positions of the four Weyl nodes in our model. The chirality of each Weyl node is presented by the plus/minus sign. **c–e** Band structure as a function of $k_x$ with $A = 0.3$, $\theta = 0$, $k_1 = 1.3\pi/2$, $k_y = 0$, and $k_z = \pi/2$. **c** Without CDW, four Weyl nodes are at the $\pm (1.3\pi/2, 0, \pm \pi/2)$ with an energy difference around 0.6 eV. **d** Without CDW ($\delta = 0$), the folded bands in the double supercell BZ along the x-direction. Weyl nodes with opposite chirality are nested out of each other's

dispersion cone. The vertical blue lines indicate the boundary of the reduced BZ along $k_x$, as used consistently throughout the paper. **e** With CDW, CDW Q-vector is along $(\pi, 0, 0)$ and the CDW strength $\delta = 0.1$. The Weyl nodes cannot be gapped and the system remains in the semimetal phase. **f–h** Band structure along the $k_x$ with $A = 0.3$, $k_1 = 1.1\pi/2$, $k_y = 0$, and $k_z = \pi/2$. **f** Without CDW, four Weyl nodes are at the $\pm (1.1\pi/2, 0, \pm \pi/2)$. **g** Without CDW ($\delta = 0$), the folded bands in the double supercell BZ along the x-direction. Weyl nodes with opposite chirality are nested into each other's dispersion cones. **h** With CDW, CDW Q-vector is along $(\pi, 0, 0)$ and the CDW strength $\delta = 0.1$. A global gap was opened by the CDW around Fermi energy.

To be more representative of real materials[22,23,26–28], we discuss the general case where the two interacting Weyl nodes lie at different energies ($E_{W_1} \neq E_{W_2}$). We consider an illustrative example using $A = 0.3$ and $k_1 = 1.3\pi/2$, which yields four Weyl nodes at $\pm (1.3\pi/2, 0, \pm \pi/2)$ with an energy difference of around 0.6 eV (Fig. 1c). To understand how the Weyl nodes are folded in the double supercell BZ (the reduced BZ), the folded band structure for the double supercell along the $x$-direction with CDW interaction strength $\delta$ set as 0 is plotted in Fig. 1d. The Weyl nodes are folded into the outside of the dispersion cone of each other. For nonzero CDW interaction strength $\delta = 0.1$, we find that the Weyl nodes do not annihilate each other and the system remains in the metallic phase (Fig. 1e). To figure out the condition for two Weyl nodes to annihilate when $E_{W_1} \neq E_{W_2}$ and $\mathbf{Q}_{CDW} \neq \mathbf{k}_{W_1} - \mathbf{k}_{W_2}$, we change the location of Weyl nodes to $\pm (1.1\pi/2, 0, \pm \pi/2)$ (Fig. 1f). In this case, the Weyl nodes in the folded BZ are within the dispersion cone of each other (Fig. 1g). Surprisingly, after the inclusion of a nonzero CDW interaction strength $\delta$, a global gap between the conduction and valence bands is seen to open up (Fig. 1h).

### Energy-momentum analog of causal structure

We find that whether or not the Weyl-CDW pairing interaction will drive the topological semimetal-insulator phase transition depends on the relative location of the Weyl nodes in the reduced BZ. In analogy with the causal structure in the theory of relativity, we define the region in energy-momentum space within (outside) the Weyl cone as the energy-like (momentum-like) region (Fig. 2a). We can express these energy-like and momentum-like regions (Figs. 2b and 2c, left panels) as

$$\begin{cases} (\delta E / V_F)^2 - (\delta \mathbf{k})^2 < 0 \;:\; \text{Momentum-like} \\ (\delta E / V_F)^2 - (\delta \mathbf{k})^2 > 0 \;:\; \text{Energy-like} \end{cases}, \qquad (2)$$

where $\delta \mathbf{k} = \mathbf{k}'_{W_1} - \mathbf{k}'_{W_2}$, and $\mathbf{k}'_{W_1}$ and $\mathbf{k}'_{W_2}$ are the new Weyl nodes positions in the reduced BZ, $\delta E = E'_{W_1} - E'_{W_2}$ is the energy difference of two

Weyl nodes with the CDW interaction, and $V_F$ is the Fermi velocity of the Weyl cone. Here we used the sign convention of the Minkowski metric $\eta_{\mu\nu} = \text{diag}(1, -1, -1, -1)$. For simplicity, we first assume that the two Weyl fermions have the same Fermi velocity, and that the Fermi velocity is isotropic.

Within the aforementioned classification, we find that the Weyl semimetal can become an insulator only when the pair of Weyl nodes around the Fermi level with CDW interaction are energy-like (Fig. 2b, right panel). Note that, in this scenario, the Weyl nodes may not be gapped, but the whole system has a semimetal to insulator transition. In contrast, if the two Weyl nodes are momentum-like, the Weyl system remains in the semimetal phase even after the CDW phase transition (Fig. 2c, right panel). Whether or not a system can undergo the metal-to-insulator transition is thus equivalent to examining whether the Weyl nodes with opposite charges are energy-like in the CDW phase.

We turn now to briefly discuss the relationship between the theory of relativity and our Weyl-CDW physical picture. In Einstein's theory of relativity, space and time are connected by the speed of light and cannot be described independently. Causality means that a cause cannot have a causal connection (effect) on an observer if it does not lie in the light cone of the observer. That is, two events can be causally related only when they are time-like. A horizon is a boundary in spacetime beyond which events cannot affect an observer. Similarly, for a system with Weyl fermions, energy-momentum space can be viewed as the analog of spacetime with Fermi velocity playing the role of speed of light. In the energy-momentum space, only when the two Weyl nodes are energy-like can they have correlation (causal connection) and make the system undergo a phase transition, and the Weyl cone plays the role of the horizon in the energy-momentum space. Note that, although there is some existing literature relating Einstein's theory of relativity to Weyl materials[29–31], the earlier work considers only a single Weyl node, while our focus is on the causal structure of the energy-momentum space and interacting Weyl systems, which requires at least two Weyl nodes.

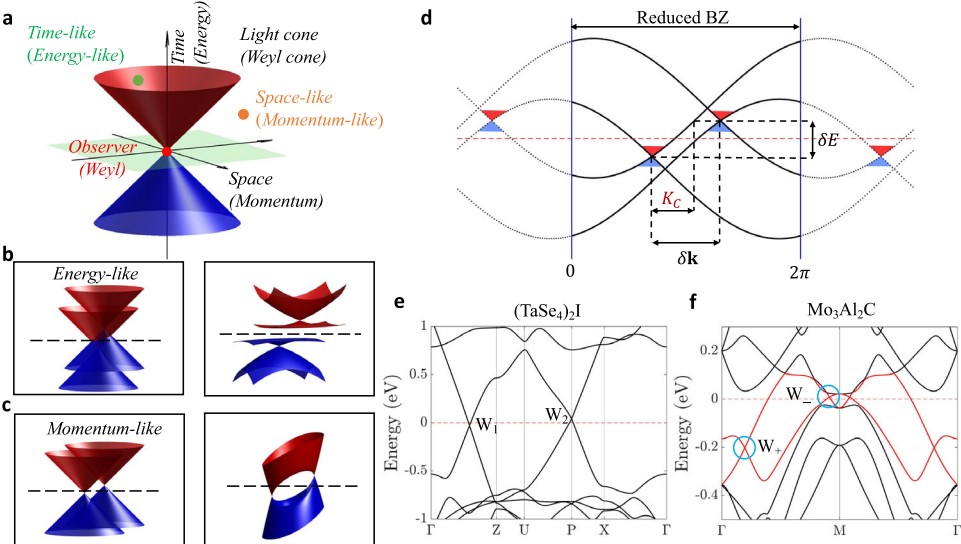

**Fig. 2 | Causal structure in energy-momentum space and its applications. a** In analogy with the theory of relativity, we define the region in energy-momentum space within/outside the Weyl cone as the energy-like/momentum-like region. **b** In the left panel, two Weyl nodes are energy-like in the reduced BZ. In the right panel, including interaction, the system undergoes a semimetal-insulator phase transition. **c** In the left panel, two Weyl nodes are momentum-like in the reduced BZ. In the right panel, including interaction, the system remains in a semimetal phase. **a–c** The red/blue structures represent the conduction/valence dispersion cones. The

black dashed lines in (**b**, **c**) represent the position of Fermi energy. **d** Schematic picture of critical length $K_C$ in the reduced BZ. $\delta \mathbf{k}$ and $\delta E$ represent the momentum and energy differences, respectively, between two Weyl nodes after band-folding without the CDW ($\delta = 0$). **e** Band structure of $(TaSe_4)_2I$ from the tight-binding model. A pair of Weyl nodes, $W_1$ and $W_2$, cross the Fermi energy. **f** The DFT band structure of $Mo_3Al_2C$ shows two bands (represented by the red curves) with a pair of Weyl nodes, $W_-$ and $W_+$ (highlighted in the blue circles), crossing the Fermi level along the Γ-M direction.

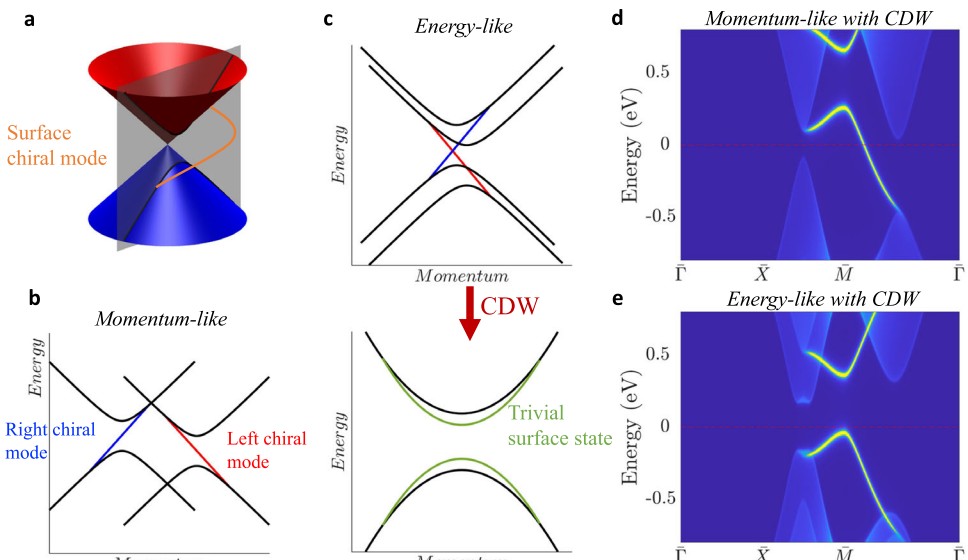

**Fig. 3 | Causality of the surface states. a** A path away from the Weyl node with the surface chiral mode connecting the conduction and valence bands. **b** An illustration of how the surface chiral modes do not cross each other in the momentum-like case. The red (blue) curves in (**b, c**) represent the left (right) surface chiral mode. **c** An illustration of how the surface chiral modes cross each other (top panel) and become trivial surface states including the CDW (bottom panel) in the energy-like case. **d** The [010] surface state of Fig. 1e. The topological surface states (yellow bright curves) connect the bulk (blur regions) conduction bands to the valance bands. **e** The [010] surface state of Fig. 1h. The surface states become topologically trivial and open a global gap in the surface band structure.

For practical purposes, we further simplify Eq. (2) to consider the case when the CDW interaction can be treated as a perturbation. We can assume that the energy and momentum of the Weyl point only acquire a small correction from CDW interaction in its reduced BZ. We define a critical length in the energy-momentum space based on the energy difference of Weyl points and their Fermi velocities as $K_C = |(E_{W_1} - E_{W_2})/V_F|$. Hence, we can ascertain the possibility of a metal-insulator transition in the Weyl-CDW system by simply comparing the critical length $K_C$ and the length of momentum separation after band-folding without the CDW ($\delta = 0$)(Fig. 2d).

**Application to real materials**
Because Weyl fermions are quite common in inversion-symmetry-breaking systems[32,33], our theory can be widely applied to the large class of noncentrosymmetric CDW materials. As an example, we consider $(TaSe_4)_2I$, which is a Weyl semimetal at room temperature that turns into an incommensurate CDW phase with $\mathbf{Q}_{CDW} = (0.027(\frac{2\pi}{a}), 0.027(\frac{2\pi}{a}), 0.012(\frac{2\pi}{c}))$ for temperatures below to 263 K[22,23]. The Weyl nodes of $(TaSe_4)_2I$ (without SOC) are shown in Fig. 2e: The energy difference is $\delta E = 0.068$ eV and the fermi velocity is $V_F \sim 3.47$ eV·Å. Thus $K_C$ is estimated to be around 0.02 Å$^{-1}$. We take approximate the CDW supercell to the nearest rational number as a commensurate supercell. Based on the folded band structure in a $37\sqrt{2} \times 37\sqrt{2} \times 83$ commensurate supercell, we find the momentum difference between these two nodes to be around 0.009 Å$^{-1}$, which is much smaller than $K_C$. Therefore, with the inclusion of CDW, the pair of Weyl nodes in $(TaSe_4)_2I$ system is energy-like and the system will become an insulator. This is consistent with experimental measurements[23].

Our arguments bear on understanding the origin of the CDW in $Mo_3Al_2C$[34,35], where it has been argued that the sudden change in the electronic density of states is due to Fermi surface nesting along the CDW nesting vector along the $(1,1,1)$ direction[34]. However, it is also reported that there is no sign of the semimetal-to-insulator transition in the $Mo_3Al_2C$[35]. Here, we examine the band structure of $Mo_3Al_2C$ without including SOC. We find a pair of Weyl nodes along the $(1,1,0)$ ($\Gamma$-M) direction in the bands that cross the Fermi energy (Fig. 2f), with

the separation of the folded Weyl nodes ($\sim 0.23$ Å$^{-1}$) being larger than $K_C \sim 0.17$ Å$^{-1}$. Therefore, the pair of Weyl nodes in $Mo_3Al_2C$ would be momentum-like with CDW interaction. These results allow us to conclude that the CDW in $Mo_3Al_2C$ leads to partial Fermi surface gapping but not to a metal-insulator transition.

**Entangled causality between bulk and surface**
We discuss the causal structure of the topological surface states in correlated Weyl semimetals by considering a path cut through the Weyl cone (away from the node) where we have a surface chiral mode connecting the gapped conduction and valence bands (Fig. 3a). The topological chiral modes exhibit opposite directions for Weyl cones of different signs of the Chern numbers. In the momentum-like case, the two surface chiral modes cannot cross (Fig. 3b). After the CDW interaction is included, there is no causal interaction between the chiral modes with opposite chirality. Thus, the surface states in the momentum-like case remain as chiral modes connecting the conduction and valance bands. In contrast, in the energy-like case, the surface chiral modes with opposite chirality cross each other (Fig. 3c, top panel). As a result, in the presence of the CDW, these chiral modes can interact to open up a surface bandgap (Fig. 3c, bottom panel). Here, we show the [010] surface states under CDW interaction using the iterative green function method[36] for momentum-like (Fig. 3d) and energy-like (Fig. 3e) cases. As we can see, the causality on the surface matches that in the bulk which presents entangled causality between the bulk and surface states in topological materials. This indicates that the causal structure of two interacting Weyl quasiparticles can be determined by observing the behavior of the chiral edge states without knowing the causality in the bulk.

**Causal structures with quantum horizon region**
In the theory of relativity, the speed of light is a universal constant. However, in our condensed matter analog, the Fermi velocities of Weyl fermions can be different from each other and are not constrained to be the speed of light. Accordingly, we now consider the case where the two Weyl fermions have different Fermi velocities by choosing non-zero $\theta$ values in Eq. (1) (Fig. 4a). We define $V_{F,H}$ ($V_{F,L}$) as the higher

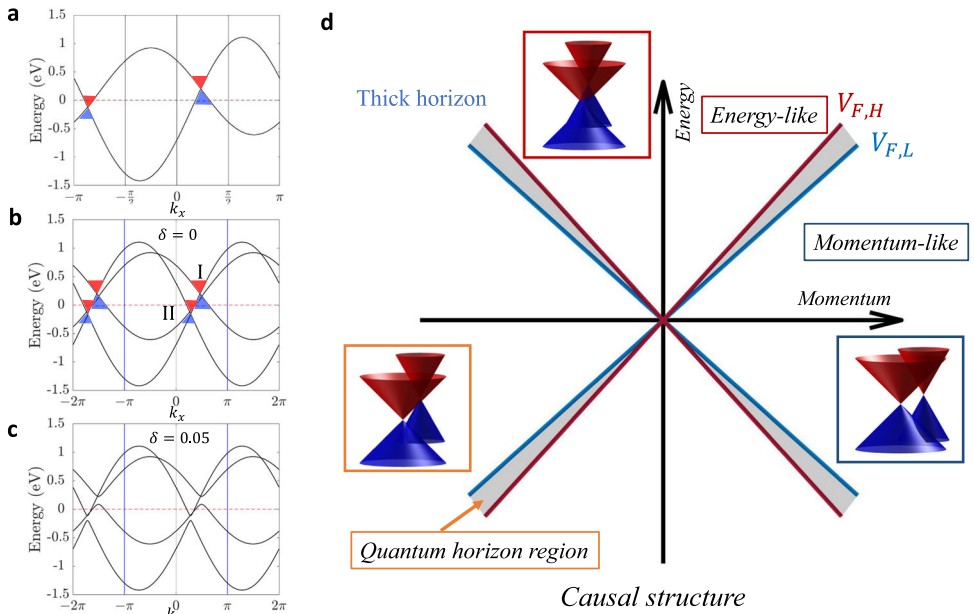

**Fig. 4 | Causal structures with quantum horizon region. a–c** Band structure as a function of $k_x$ with $A = 0.3$, $\theta = 1$, $k_1 = 1.1\pi/2$, $k_y = 0$, and $k_z = \pi/2$. **a** Without CDW, the Fermi velocities of the two Weyl quasiparticles are different. **b** Without CDW ($\delta = 0$), the folded bands in the double supercell BZ along the x-direction. The Weyl node I is in the energy-like region of Weyl node II but Weyl node II is in the momentum-like region of Weyl node I. **c** With CDW of Q-vector along $(\pi, 0, 0)$ and the CDW strength $\delta = 0.05$: The Weyl nodes remain intact and there is no metal-insulator phase transition. **d** The causal structure of a correlated Weyl system. The quantum horizon region is shown as a "thick horizon" bounded by the Fermi velocities $V_{F,H}$ and $V_{F,L}$ of the two interacting fermions.

(lower) Fermi velocity of the two Weyl fermions. The gapping condition is found to remain unchanged when the two Weyl fermions lie inside or outside each other's dispersion cone. Specifically, when the two Weyl nodes lie in the energy-like regions $[(\delta E/V_{F,H})^2 - (\delta \mathbf{k})^2 > 0]$, the whole system is gapped out by the CDW interaction. In contrast, when the two Weyl nodes lie in the momentum-like regions $[(\delta E/V_{F,L})^2 - (\delta \mathbf{k})^2 < 0]$, the system remains gapless.

Because of the different Fermi velocities involved, a unique phase can emerge in our system in which the Weyl node I lies in the energy-like region of the Weyl node II but the Weyl node II lies in the momentum-like region of Weyl node I (Fig. 4b). When a small non-zero CDW interaction is included, the Weyl nodes remain intact and no global band gap is seen (Fig. 4c). However, the system is near a quantum critical point and a slight increase in the strength of the interaction can drive the two Weyl fermions to fall into each other's dispersion cone (become energy-like) and open a global band gap, see Supplementary Fig. 3. Interestingly, the causal structure of the interacting Weyl system could thus be changed by tuning the strength of the interaction. These results open-up opportunities for exploring causal structures beyond the framework of Einstein's theory of relativity in which space-like to energy-like crossover is forbidden. Since we are close to a quantum critical point, we refer to this region as the "quantum horizon region" that bridges energy-like and momentum-like regions.

We summarize the full causal structure of the interacting Weyl system in Fig. 4d where the quantum horizon region is shown as a 'thick horizon' bounded by $V_{F,H}$ and $V_{F,L}$ via the equations:

$$\begin{cases} (\delta E/V_{F,L})^2 - (\delta \mathbf{k})^2 < 0 : \text{Momentum-like} \\ (\delta E/V_{F,H})^2 - (\delta \mathbf{k})^2 > 0 : \text{Energy-like} \\ |V_{F,H}| \geq |\delta E/\delta \mathbf{k}| \geq |V_{F,L}| : \text{Quantum horizon region} \end{cases} \quad (3)$$

Based on this causal structure, we comment on the reason why the CDW interaction is always attractive in the sense that interaction moves the two Weyl quasiparticles toward each other. Recall that the CDW interaction tends to open a global band gap and drives the system into an insulator phase. However, in a Weyl system, a global band gap between the bands forming the Weyl nodes can only be opened by annihilating the two interacting Weyl quasiparticles. Since causal interaction is only possible between energy-like Weyl quasiparticles, the CDW interaction can be expected to move the two Weyl quasiparticles closer in the energy-momentum space (Supplementary Figs. 2, 3). Therefore, it is only possible to have a crossover from the quantum horizon region to the energy-like region but not to the momentum-like region with increasing interaction strength.

When considering the high-energy analogs of Weyl fermions in condensed matter systems, Weyl cone is often identified as a well-defined relativistic quasiparticle by using a proper linear approximation for its dispersion cone around the node. However, although the non-linear dispersion terms violate the Lorentz invariance in high-energy physics, it is natural for condensed matter systems that the dispersion cone of the Weyl node becomes non-linear due to quadratic and higher-order corrections. Due to the non-linear dispersion, it's also natural that the two Weyl nodes may not both lie inside (outside) of each other's dispersion cone and thus impact the emergence of the quantum horizon region in the causal structure, see Supplementary Fig. 4. The non-linear dispersion cones indicate that the Eq. (3) can be further generalized to the case simply based on whether the two interacting Weyl quasiparticles are inside or outside of each other's dispersion cone as follows:

$$\begin{cases} \text{Both outside} : \text{Momentum-like} \\ \text{Both inside} : \text{Energy-like} \\ \text{One inside, one outside} : \text{Quantum horizon region} \end{cases} \quad (4)$$

Based on our generalized conclusions for the Lorentz violating cases, we conjecture that the casual structure of interacting type-II Weyl quasiparticles[37], where the Lorentz invariance is violated and the Fermi velocities for the two branches of the dispersion cones possess the

same sign, criteria similar to those in Eq. (4) would be applicable. Note that there is a connection between the behavior of the energy spectrum behind the event horizon of a blackhole and the type-II Weyl fermions[30], suggesting that the causal structure of the interacting type-II Weyl fermions may hide yet more profound new physics that would be interesting to explore. Finally, we note that the causal structure shown in Eq. (3) and Eq. (4) can also be derived from the low-energy effective **k·p** description, so that our gap-opening condition is universal and independent of the choice of the model and the form of the interaction, see the Supplementary Fig. 5–6.

## Discussion

In analogy with the spacetime light cone and the related causality-driven event horizon in relativistic physics, we have unveiled the causal structure of the energy-momentum space in the condensed matter context and show that it consists of energy-like, momentum-like, and quantum horizon regions. Our analysis reveals that a correlated Weyl system can realize a topological metal-insulator transition only when a pair of interacting Weyl nodes with opposite topological charges are energy-like, otherwise they are forbidden to interact to produce a band gap. We also demonstrate that only when the interacting Weyl fermions are energy-like that the two opposite chiral surface modes can have a causal connection. In this sense, the quantum information (causality) stored in a volume element is thus also encoded on its surface, much like in the case of quantum black holes, where the quantum state outside a black hole horizon carries information about the internal state of the black hole. This result points to an interesting connection between the interacting Weyl systems and quantum black holes. Finally, our study indicates the presence of a quantum horizon region as a thick horizon in the causal structure of the interacting Weyl system.

Although we have focused on Weyl-CDW systems, our results are applicable more generally to interacting Weyl fermions in topological systems. For example, Weyl physics can be simulated in a 3D optical lattice[38], where the CDW effect could be produced by introducing a period-2 superlattice (dimerization) using two additional orthogonal optical waves at double the in-plane wavelengths[39]. In our case, the CDW is generated via the spontaneous breaking of the translational symmetry, which leads to a pairing interaction between the Weyl fermions. In view of the universality of Weyl physics, however, our formalism will apply to more general pairing interactions that break the symmetries of the system. Our study, for the first time, shows how the key concepts of causality and the associated event horizon in spacetime can be carried over into the field of correlated Weyl materials, and thus unveils fundamental connections between condensed matter and high-energy physics.

Causality was long considered as the time-ordered relationship between causes and effects until the advent of Einstein's theory of relativity, in which causality is defined by the light cone in spacetime. We have introduced causal structure in the energy-momentum space of the condensed matter systems. We can expect a far richer tapestry of possibilities driven by causal physics in the context of condensed matter systems, just as many more exotic fermionic excitations are supported by the vacuum of crystalline materials compared to that of free space.

## Methods

### CDW tight-binding model in real space

Consider a cubic system with one atom per unit cell and lattice parameter $a$. We first construct a two-band model with four Weyl points with broken inversion symmetry,

$$\mathcal{H}(k) = A \sin(k_x a) \sin(k_z a) \sigma_0 + \left[ \cos(k_x a + \theta) - \cos(k_1 a) \right] \sigma_x + \sin(k_y a) \sigma_y + \left[ 1 + \cos(k_z a) - \cos(k_y a) \right] \sigma_z. \quad (5)$$

By taking Fourier transform of the lattice tight-binding model in momentum space, we get the hopping parameters as following,

$$t_{ij} = \frac{1}{N} \sum_{\mathbf{k}}^{BZ} e^{-i\mathbf{k}\cdot\mathbf{r}} \mathcal{H}(\mathbf{k}) \quad (6)$$

$$= A\left\{ \frac{1}{2i} \left[ \delta(\mathbf{r} + a\hat{x}) - \delta(\mathbf{r} - a\hat{x}) \right] \frac{1}{2i} \left[ \delta(\mathbf{r} + a\hat{z}) - \delta(\mathbf{r} - a\hat{z}) \right] \right\} \sigma_0$$
$$+ \left\{ \frac{1}{2} \left[ e^{-i\theta}\delta(\mathbf{r} + a\hat{x}) + e^{i\theta}\delta(\mathbf{r} - a\hat{x}) \right] - \delta(\mathbf{r}) \cos k_1 \right\} \sigma_x$$
$$+ \left\{ \frac{1}{2i} \left[ \delta(\mathbf{r} + a\hat{y}) - \delta(\mathbf{r} - a\hat{y}) \right] \right\} \sigma_y$$
$$+ \left\{ \delta(\mathbf{r}) + \frac{1}{2} \left[ \delta(\mathbf{r} + a\hat{z}) + \delta(\mathbf{r} - a\hat{z}) \right] - \frac{1}{2} \left[ \delta(\mathbf{r} + a\hat{y}) + \delta(\mathbf{r} - a\hat{y}) \right] \right\} \sigma_z, \quad (7)$$

where $\mathbf{r} = \mathbf{r}_i - \mathbf{r}_j$ and $\mathbf{r}_i$, $\mathbf{r}_j$ are the displacements of lattice sites, and $\sigma$ is the Pauli matrix which describes the orbital degree of freedom on the atom. Let $a = 1$ and the two-band Hamiltonian without interaction term is

$$H_0 = \sum_{i,j} c_i^\dagger t_{ij} c_j + \text{H.c.} \quad (8)$$

$$= \sum_i \frac{-A}{4} \left[ c_i^\dagger \sigma_0 c_{i+\hat{x}+\hat{z}} + c_i^\dagger \sigma_0 c_{i-\hat{x}-\hat{z}} - c_i^\dagger \sigma_0 c_{i+\hat{x}-\hat{z}} - c_i^\dagger \sigma_0 c_{i-\hat{x}+\hat{z}} \right]$$
$$+ \sum_i \left[ -\cos k_1 c_i^\dagger \sigma_x c_i + \frac{1}{2} e^{i\theta} c_i^\dagger \sigma_x c_{i+\hat{x}} + \frac{1}{2} e^{-i\theta} c_i^\dagger \sigma_x c_{i-\hat{x}} \right]$$
$$+ \sum_i \frac{1}{2i} \left[ c_i^\dagger \sigma_y c_{i+\hat{y}} - c_i^\dagger \sigma_y c_{i-\hat{y}} \right]$$
$$+ \sum_i \left[ c_i^\dagger \sigma_z c_i + \frac{1}{2} c_i^\dagger \sigma_z c_{i+\hat{z}} + \frac{1}{2} c_i^\dagger \sigma_z c_{i-\hat{z}} - \frac{1}{2} c_i^\dagger \sigma_z c_{i+\hat{y}} - \frac{1}{2} c_i^\dagger \sigma_z c_{i-\hat{y}} \right] + \text{H.c.} \quad (9)$$

where $c_i = (c_{i,1}, c_{i,2})$ and $c_{i,1}$, $c_{i,2}$ are the electron annihilation operators with the orbital (pseudo-spin) index 1, 2 on the atom at the site $\mathbf{r}_i$. Then we consider the semi-1D CDW as Peierls dimerization in a double supercell along the $x$ direction. We build a double-cell supercell along the $\hat{x}$ direction and we denote the electron annihilation operators of the two atoms within the supercell at $\mathbf{r}'_i$ as $c_i$ and $d_i$, where $c_i = (c_{i,1}, c_{i,2})$, $d_i = (d_{i,1}, d_{i,2})$ and $\mathbf{r}'_i$ as the displacements of super lattice sites. We can write the Hamiltonian on the supercell basis as

$$H_{0,SC} = \sum_i \frac{-A}{4} \left[ c_i^\dagger \tau_0 d_{i+\hat{z}} + c_i^\dagger \tau_0 d_{i-\hat{x}-\hat{z}} - c_i^\dagger \tau_0 d_{i-\hat{z}} - c_i^\dagger \tau_0 d_{i-\hat{x}+\hat{z}} \right.$$
$$\left. + d_i^\dagger \tau_0 c_{i+\hat{x}+\hat{z}} + d_i^\dagger \tau_0 c_{i-\hat{z}} - d_i^\dagger \tau_0 c_{i+\hat{x}-\hat{z}} - d_i^\dagger \tau_0 c_{i+\hat{z}} \right]$$
$$+ \sum_i \left[ -\cos k_1 (c_i^\dagger \sigma_x c_i + d_i^\dagger \sigma_x d_i) + \frac{1}{2} e^{i\theta} c_i^\dagger \tau_x d_i + \frac{1}{2} e^{-i\theta} c_i^\dagger \tau_x d_{i-\hat{x}} \right.$$
$$\left. + \frac{1}{2} e^{i\theta} d_i^\dagger \tau_x c_{i+\hat{x}} + \frac{1}{2} e^{-i\theta} d_i^\dagger \tau_x c_i \right]$$
$$+ \sum_i \frac{1}{2i} \left[ (c_i^\dagger \sigma_y c_{i+\hat{y}} - c_i^\dagger \sigma_y c_{i-\hat{y}}) + (d_i^\dagger \sigma_y d_{i+\hat{y}} - d_i^\dagger \sigma_y d_{i-\hat{y}}) \right]$$
$$+ \sum_i \left[ c_i^\dagger \sigma_z c_i + \frac{1}{2} c_i^\dagger \sigma_z c_{i+\hat{z}} + \frac{1}{2} c_i^\dagger \sigma_z c_{i-\hat{z}} - \frac{1}{2} c_i^\dagger \sigma_z c_{i+\hat{y}} - \frac{1}{2} c_i^\dagger \sigma_z c_{i-\hat{y}} \right.$$
$$\left. + d_i^\dagger \sigma_z d_i + \frac{1}{2} d_i^\dagger \sigma_z d_{i+\hat{z}} + \frac{1}{2} d_i^\dagger \sigma_z d_{i-\hat{z}} - \frac{1}{2} d_i^\dagger \sigma_z d_{i+\hat{y}} - \frac{1}{2} d_i^\dagger \sigma_z d_{i-\hat{y}} \right]$$
$$+ \text{H.c.} \quad (10)$$

where $\tau$ is the Pauli matrix describing the degree of freedom between two atoms. Then, the Peierls dimerization can be realized by modifying the real space hopping strength between the two nearest neighbor atoms along the CDW direction in the supercell with a strength $\delta$, and

the interaction terms can be expressed in the supercell basis as

$$H_{int} = \sum_i \left[ \delta c_i^\dagger \tau_x d_i + (-\delta) c_i^\dagger \tau_x d_{i-\hat{x}} + (-\delta) d_i^\dagger \tau_x c_{i+\hat{x}} + \delta d_i^\dagger \tau_x c_i \right] + \text{H.c.} .$$

(11)

Then the full Hamiltonian becomes

$$
\begin{aligned}
H_{CDW} &= H_{0,SC} + H_{int} \\
&= \sum_i \frac{-A}{4} \left[ c_i^\dagger \tau_0 d_{i+\hat{z}} + c_i^\dagger \tau_0 d_{i-\hat{x}-\hat{z}} - c_i^\dagger \tau_0 d_{i-\hat{z}} - c_i^\dagger \tau_0 d_{i-\hat{x}+\hat{z}} \right. \\
&\quad \left. + d_i^\dagger \tau_0 c_{i+\hat{x}+\hat{z}} + d_i^\dagger \tau_0 c_{i-\hat{z}} - d_i^\dagger \tau_0 c_{i+\hat{x}-\hat{z}} - d_i^\dagger \tau_0 c_{i+\hat{z}} \right] \\
&\quad + \sum_i \left[ -\cos k_1 (c_i^\dagger \sigma_x c_i + d_i^\dagger \sigma_x d_i) + \left(\frac{1}{2} e^{i\theta} + \delta \right) c_i^\dagger \tau_x d_i + \left(\frac{1}{2} e^{-i\theta} - \delta \right) c_i^\dagger \tau_x d_{i-\hat{x}} \right. \\
&\quad \left. + \left(\frac{1}{2} e^{i\theta} - \delta \right) d_i^\dagger \tau_x c_{i+\hat{x}} + \left(\frac{1}{2} e^{-i\theta} + \delta \right) d_i^\dagger \tau_x c_i \right] \\
&\quad + \sum_i \frac{1}{2i} \left[ (c_i^\dagger \sigma_y c_{i+\hat{y}} - c_i^\dagger \sigma_y c_{i-\hat{y}}) + (d_i^\dagger \sigma_y d_{i+\hat{y}} - d_i^\dagger \sigma_y d_{i-\hat{y}}) \right] \\
&\quad + \sum_i \left[ c_i^\dagger \sigma_z c_i + \frac{1}{2} c_i^\dagger \sigma_z c_{i+\hat{z}} + \frac{1}{2} c_i^\dagger \sigma_z c_{i-\hat{z}} - \frac{1}{2} c_i^\dagger \sigma_z c_{i+\hat{y}} - \frac{1}{2} c_i^\dagger \sigma_z c_{i-\hat{y}} \right. \\
&\quad \left. + d_i^\dagger \sigma_z d_i + \frac{1}{2} d_i^\dagger \sigma_z d_{i+\hat{z}} + \frac{1}{2} d_i^\dagger \sigma_z d_{i-\hat{z}} - \frac{1}{2} d_i^\dagger \sigma_z d_{i+\hat{y}} - \frac{1}{2} d_i^\dagger \sigma_z d_{i-\hat{y}} \right] \\
&\quad + \text{H.c.}
\end{aligned}
$$

(12)

It's worth noting that the $\delta\tau_x$ terms are in the off-diagonal blocks of the four orbitals basis, which reflects the nature of Weyl nodes interacting with each other through CDW.

## First-principles calculations

First-principles calculations for $(TaSe_4)_2I$ were performed using OpenMX code, where the generalized-gradient approximation, norm-conserving pseudopotentials, and optimized pseudoatomic basis functions were adopted[40–43]. Three, two, two, and one optimized radial functions were allocated for the s, p, d, and f orbitals, respectively, for each Ta atom with a cut-off radius of 7 Bohr, denoted as Ta7.0-s3p2d2f1. For the Se and I atoms, Se7.0-s3p3d2f1 and I7.0-s3p3d2f1 were adopted, respectively. A cutoff energy of 300 Ry was used for numerical integrations and for the solution of the Poisson equation. The first-principles calculations on $Mo_3Al_2C$ were carried out using the Vienna Ab initio Simulation Package with the projector augmented wave potentials[44]. The exchange-correlation function was treated within the Perdew–Burke–Ernzerhof generalized gradient approximations[41].

## Data availability

The data that support the findings of this study are available from the corresponding authors upon request.

## Code availability

All code used to generate the plotted band structures is available from the corresponding author upon request.

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

## Acknowledgements

We thank Justin Ripley for a helpful discussion. G.C. acknowledges the support of the National Research Foundation, Singapore under its Fellowship Award (NRF-NRFF13-2021-0010) and the Nanyang Assistant Professorship grant from Nanyang Technological University. The work at Northeastern University was supported by the Air Force Office of Scientific Research under award number FA9550-20-1-0322, and it benefited from the computational resources of Northeastern University's Advanced Scientific Computation Center (ASCC) and the Discovery Cluster. M.Z.H. was supported by the US DOE under the Basic Energy Sciences program (grant number DOE/BES DE-FG-02-05ER46200). S.M.H. is supported by the NSTC-AFOSR Taiwan program on Topological and Nanostructured Materials, Grant No. 110-2124-M-110-002-MY3. C.-C.L. acknowledges the National Science and Technology Council (NSTC) of Taiwan for financial support under Contract No. 110-2112-M-032-016-MY2. F.C.C. acknowledges the support by the National Center for Theoretical Sciences and the Ministry of Science and Technology of Taiwan under grant no. MOST-110-2112-M-110-013-MY3. S.-Y.X. acknowledges the support of the Center for the Advancement of Topological Semimetals (CATS), an Energy Frontier Research Center (EFRC) funded by the US Department of Energy (DOE) Office of Science, through the Ames Laboratory under contract DE-AC0207CH11358 (fabrication and measurements), the STC Center for Integrated Quantum Materials (CIQM), National Science Foundation (NSF) award no. ECCS-2025158 (data analysis), and the Corning Fund for Faculty Development. H.L. acknowledges the support by the National Science and Technology Council (NSTC) in Taiwan under grant number MOST 111-2112-M-001-057-MY3. T.-R.C. was supported by the Young Scholar Fellowship Program from the Ministry of Science and Technology (MOST) in Taiwan, under a MOST grant for the Columbus Program MOST110-2636-M-006-016, the National Cheng Kung University, Taiwan, and National Center for Theoretical Sciences, Taiwan. Work at NCKU was supported by MOST, Taiwan, under grant MOST107-2627-E-006-001 and Higher Education Sprout Project, Ministry of Education to the Headquarters of University Advancement at NCKU.

## Author contributions

All the authors contributed to the intellectual content of this work. W.-C.C. and G.C. initiated the project. W.-C.C. proposed the conceptual idea of causal structure in energy-momentum space and performed numerical simulations and analytical calculations with assistance from G.C., R.M., and S.-M.H.. G.M. and G.C. performed first-principles calculations and analysis with assistance from C.-C.L., T.-R.C., F.-C.C, H.L., and A. B. The materials search was done by G.C. with help from G.M., I.B., J.-X.Y., Z.-J.C., C.-C.L. S.-Y.X., and M.Z.H. W.-C.C. wrote the original draft, and G.C., R.M., I.B., and A.B. revised the draft. G.C. and A.B. were responsible for the overall research direction, planning, and integration among different research units.

## Competing interests

The authors declare no competing interests.
