## [Peer Review File · Nature Communications]

Causal structure of interacting Weyl fermions in condensed matter systemsEditorial Note: This manuscript has been previously reviewed at another journal that is not operating a transparent peer review scheme. This document only contains reviewer comments and rebuttal letters for versions considered at Nature Communications.

Reviewers' Comments:

Reviewer #2:

Remarks to the Author:

In their revised manuscript, the authors have added several new analyses and results that significantly strengthen the work. In particular, the k.p analysis, the discussion of CDW amplitude effects, and the identification of a new "quantum critical" phase when Weyl cones have different Fermi velocities strengthen the current work and point towards new areas of future research. I also appreciate the emphasis in the revised manuscript on the origin of the CDW modulation as well as the analysis of surface states. I think the revised manuscript is in principle suitable for publication in Nature Communications.

That said, I have a few critiques that I hope the authors will address in a final version of the manuscript:

1. While it is true that Refs. 29-31 focus primarily on a single Weyl node, I don't think it's correct to say that they ignore the causal structure in the BZ. In particular, Ref. 30 makes the point that Type-II Weyl fermions resemble black holes in the sense that their future light cone opens in a spacelike separated direction. This is precisely the sort of statement about causality that should have profound implications in the formalism of the present manuscript. While it is certainly not necessary for the authors to consider every permutation of possible Weyl-CDW band structures in the present work, I think it would greatly improve the manuscript if the authors could comment or speculate on how the light cone structure of type-II Weyl fermions could affect the present analysis (i.e., should type-II Weyl fermions that are momentum-like separated after modulation behave more like type-I Weyl fermions that are energy-like separated?). In that way, the authors would contextualize their work on multi-Weyl causal structures into the known results on single-Weyl causal structures.
2. I would caution the authors on the use of the phrase "quantum critical region," as this is used in other context to refer to the region of a phase diagram at finite temperature near a quantum critical point. While its clear why the authors use this phrase, and its technically correct, it could cause confusion in a journal with broad readership like Nature Communications (unless of course, the authors would like to comment on the effect of finite temperature in the critical region).

Reviewer #3:

Remarks to the Author:

I have read the response from the authors and I find they have not avoided my questions and have very explicitly addressed all my concerns by adding more substantial calculations and discussions in the new version. With these new ingredients, I think the present version is satisfactory and reaches high rigor, which is comparable to similar topics of research articles in Nature journals.

Reviewer #4:

Remarks to the Author:

I was asked to review this article as a causality expert. However, my work is about going beyond causality and exploring how quantum mechanics might allow probing these situations. The paper uses the standard causality paradigm, with light cones and a flat spacetime, so I don't think there's anything wrong with that. However, the point they are trying to make is beyond my expertise. Hence, I would rely on the two previous reviewers who seem to be more appropriate in reviewing this manuscript.

AUTHORS' RESPONSE

We sincerely thank the reviewers for taking the time to study our manuscript carefully and for their constructive suggestions. We address the few remaining questions raised by the reviewers in point-by-point detail below. The manuscript has also been revised accordingly. We have highlighted the revisions in the manuscript with a blue font for the convenience of the reviewers.

Response to Reviewer 2

In their revised manuscript, the authors have added several new analyses and results that significantly strengthen the work. In particular, the $k.p$ analysis, the discussion of CDW amplitude effects, and the identification of a new "quantum critical" phase when Weyl cones have different Fermi velocities strengthen the current work and point towards new areas of future research. I also appreciate the emphasis in the revised manuscript on the origin of the CDW modulation as well as the analysis of surface states. I think the revised manuscript is in principle suitable for publication in Nature Communications. That said, I have a few critiques that I hope the authors will address in a final version of the manuscript:

Authors Response: We sincerely appreciate the time and effort taken by the reviewer in meticulously reviewing our work. We would like to express our heartfelt thanks to the reviewer for agreeing that we have “*significantly strengthened the work*” in our revised manuscript. We are grateful to the reviewer for recognizing that our revised manuscript “*point towards new areas of future research*” and is “*in principle suitable for publication in Nature Communications*”.

Reviewer #2: Point 1: *While it is true that Refs. 29-31 focus primarily on a single Weyl node, I don't think it's correct to say that they ignore the causal structure in the BZ. In particular, Ref. 30 makes the point that Type-II Weyl fermions resemble black holes in the sense that their future light cone opens in a spacelike separated direction. This is precisely the sort of statement about causality that should have profound implications in the formalism of the present manuscript. While it is certainly not necessary for the authors to consider every permutation of possible Weyl-CDW band structures in the present work, I think it would greatly improve the manuscript if the authors could comment or speculate on how the light cone structure of type-II Weyl fermions could affect the present analysis (i.e., should type-II Weyl fermions that are momentum-like separated after modulation behave more like type-I Weyl fermions that are energy-like separated?). In that way, the authors would contextualize their work on multi-Weyl causal structures into the known results on single-Weyl causal structures.*

Authors' Response: We thank the reviewer for pointing out that “*Ref. 30 makes the point that Type-II Weyl fermions resemble black holes in the sense that their future light cone opens in a spacelike separated direction.*” Accordingly, we have now removed the sentence that implied that Ref. [29-31] “*without attention to causal structure*” in the revised manuscript.

The reviewer raised a thoughtful question related to type-II Weyl quasiparticles, which indeed opens a promising direction for future work. In response to the referee’s request to “*comment or speculate on how the light cone structure of type-II Weyl fermions could affect the present analysis*”, we have added the following remarks at the end of our revised manuscript:

“Based on our generalized conclusions for the Lorentz violating cases, we conjecture that the casual structure of interacting type-II Weyl quasiparticles [Nature 527, 495–498 (2015)], where the Lorentz invariance is violated and the Fermi velocities for the two branches of the dispersion cones possess the same sign, criteria similar to those in Eq. (4) would be applicable. Note that there is a connection between the behavior of the energy spectrum behind the event horizon of a black-hole and the type-II Weyl fermions [30], suggesting that the causal structure of the interacting type-II Weyl fermions may hide yet more profound new physics that would be interesting to explore.”

Reviewer #2: Point 2: *I would caution the authors on the use of the phrase “quantum critical region,” as this is used in other context to refer to the region of a phase diagram at finite temperature near a quantum critical point. While its clear why the authors use this phrase, and its technically correct, it could cause confusion in a journal with broad readership like Nature Communications (unless of course, the authors would like to comment on the effect of finite temperature in the critical region).*

Authors' Response: We appreciate the reviewer’s cautionary remark, which provoked further thought and helped us improve our manuscript. We have addressed this concern in the revised manuscript as follows.

To avoid the confusion on the usage of the term “*quantum critical region*”, we have now replaced the term “*quantum critical region*” by “*quantum horizon region*” throughout the revised manuscript for describing the region close to the quantum critical point that bridges the energy-like and momentum-like regions.

Response to Reviewer 3

I have read the response from the authors and I find they have not avoided my questions and have very explicitly addressed all my concerns by adding more substantial calculations and discussions in the new version. With these new ingredients, I think the present version is satisfactory and reaches high rigor, which is comparable to similar topics of research articles in Nature journals.

Authors' Response: We are very grateful to the reviewer for his/her time and effort in reviewing our manuscript. We would like to express our heartfelt thanks to the reviewer for recognizing that our revised manuscript “*is satisfactory and reaches high rigor, which is comparable to similar topics of research articles in Nature journals*”. We are grateful to the reviewer for his/her strong endorsement of our study.

Response to Reviewer 4

I was asked to review this article as a causality expert. However, my work is about going beyond causality and exploring how quantum mechanics might allow probing these situations. The paper uses the standard causality paradigm, with light cones and a flat spacetime, so I don't think there's anything wrong with that. However, the point they are trying to make is beyond my expertise. Hence, I would rely on the two previous reviewers who seem to be more appropriate in reviewing this manuscript.

Authors' Response: We are grateful to the reviewer for taking the time to study our manuscript and recognizing that there's nothing wrong with our theoretical framework from his/her perspective of causality.

Reviewers' Comments:

Reviewer #2:

Remarks to the Author:

The authors have attentively addressed all of my remaining concerns. I recommend publication.